# Sexually Dimorphic Gene Expression in X and Y Sperms Instructs Sexual Dimorphism of Embryonic Genome Activation in Yellow Catfish (*Pelteobagrus fulvidraco*)

**DOI:** 10.3390/biology11121818

**Published:** 2022-12-14

**Authors:** Yang Xiong, Dan-Yang Wang, Wenjie Guo, Gaorui Gong, Zhen-Xia Chen, Qin Tang, Jie Mei

**Affiliations:** 1College of Fisheries, Huazhong Agricultural University, Wuhan 430070, China; 2College of Biomedicine and Health, Huazhong Agricultural University, Wuhan 430070, China; 3Hubei Hongshan Laboratory, Wuhan 430070, China

**Keywords:** sex instructor, sexual dimorphism, MZT, gene expression, DNA methylation, sex determination

## Abstract

**Simple Summary:**

After fertilization, maternal degradation, zygotic activation, and paternal activation are three major activities during early embryonic development. Genes involved in these activities are assumed to contribute to sex determination and sexual dimorphism. To our knowledge, paternal genes are activated along with zygotic gene activation, but the influence of paternal inheritance on sex determination and sexual dimorphism is still unclear. As a potential model, yellow catfish (*Pelteobagrus fulvidraco*) is sexually dimorphic in growth rate with males growing faster than females. We performed sex-control breeding of this fish and successfully created XX neo-male and YY super-males by sex reversal technology, which provided purified X and Y spermatozoa for paternal genetic research. Studies of gene expression and DNA methylation of spermatozoa indicated a big proportion of upregulated genes in Y sperm with slightly lower DNA methylation levels. Maternal-to-zygotic transition (MZT) processes were identified from the XX female and XY male offspring through a combined analysis of SNP and transcriptomic dynamics, demonstrating the low-blastocyst stage as a critical time point of zygotic and paternal activation. The emergence of sex differences lagged behind gene expression differences. Integration analysis of X/Y sperm and embryos revealed the influence of paternal inheritance on sexual dimorphism of genome activation.

**Abstract:**

Paternal factors play an important role in embryonic morphogenesis and contribute to sexual dimorphism in development. To assess the effect of paternal DNA on sexual dimorphism of embryonic genome activation, we compared X and Y sperm and different sexes of embryos before sex determination. Through transcriptome sequencing (RNA-seq) and whole-genome bisulfite sequencing (WGBS) of X and Y sperm, we found a big proportion of upregulated genes in Y sperm, supported by the observation that genome-wide DNA methylation level is slightly lower than in X sperm. Cytokine–cytokine receptor interaction, TGF-beta, and toll-like receptor pathways play important roles in spermatogenesis. Through whole-genome re-sequencing (WGRS) of parental fish and RNA-seq of five early embryonic stages, we found the low-blastocyst time point is a key to maternal transcriptome degradation and zygotic genome activation. Generally, sexual differences emerged from the bud stage. Moreover, through integrated analysis of paternal SNPs and gene expression, we evaluated the influence of paternal inheritance on sexual dimorphism of genome activation. Besides, we screened out *gata6* and *ddx5* as potential instructors for early sex determination and gonad development in yellow catfish. This work is meaningful for revealing the molecular mechanisms of sex determination and sexual dimorphism of fish species.

## 1. Introduction

The XX/XY sex-determination system exists in most mammals and many reptiles and fishes, in which XY males produce X and Y sperms. Although some morphological differences between human X and Y spermatozoa have been reported in some earlier preliminary studies using phase-contrast microscopy [1,2,3], they have been refuted by some recent human and bovine studies using more specific technologies [4,5,6]. In addition, controversial findings over the physiological index of X and Y spermatozoa, including sperm motility, viability, and sensitivity to environmental factors have been revealed between X and Y spermatozoa [7]. To date, very limited studies have characterized differentially expressed genes and proteins from sorted X and Y spermatozoa that do not have high-purity [8,9,10,11]. Therefore, the differences between X and Y spermatozoa are still uncertain due to the lack of accurate methods to discriminate the two types of spermatozoa.

Sex determination and sex differentiation in vertebrates have been extensively investigated [12,13,14,15]. After fertilization, the chromosome combination of X oocyte with X or Y sperm will create either a female embryo (XX) or a male embryo (XY). Typically, the early gonads of XX or XY individuals are undifferentiated gonads that develop into testes or ovaries after sex determination [15]. However, dynamic gene expression between XX and XY embryos is unknown before sex determination. During early embryogenesis, the first developmental event in vertebrate embryos is the maternal-to-zygotic transition (MZT), when maternally deposited mRNAs are degraded and zygotic transcription is activated [16]. Meantime, paternal genes are activated along with the zygotic activation. At the early time of embryonic development, the primordial germ cells (PGCs), a cluster of cells expressing sex-determining genes, also began to form [17]. PGCs migrate towards the genital ridge and differentiate into gametes after their production, affecting the formation of gonads (the testes and ovaries) [18]. Many maternal mRNAs are essential for the PGC specification and differentiation; loss of their functions will affect sex differentiation [19,20,21]. Germplasm genes such as *vasa*, *dazl*, *buc*, and *nanos*, could affect the specification of PGC [22]. The sex-determining gene, *wasp overruler of masculinization* (*wom*), which is paternally transcribed from the fertilized eggs, is also an instructor of PGC, and can instruct female sex determination in the haplodiploid wasp *Nasonia* [23]. Thus, there must exist some specific instructor signals that guide and start the sex-determination pathway during early embryo development. Previously, we have reported three types of master genes, including DM (doublesex and Mab-3) domain-containing genes, TGF-β signaling pathway genes, and *sox* family genes involved in the sexual regulatory network [24]. However, it is not clear whether more signals mediate the interaction between sex-determining related pathways in the network.

Sex-biased gene expression is driven by sex-chromosome-based transcription, and sperm is one of the original carriers of sex chromosomes [25]. Recently, we explored differences between the sex chromosomes of yellow catfish from an evolutionary perspective and revealed about 1% nucleotide divergence between X and Y [26]. However, the extent to which X and Y sperm divergence contribute to the sexual dimorphism of offspring still needs to be revealed. Except for maternal and zygotic genes, paternal genes from sperms are also assumed to contribute to sex determination and dimorphism. Sex determination is a highly plastic process in lower vertebrates including reptiles, amphibians and fishes [12,15,27], which provides us with chances to perform sex-controlling breeding and reveal the mechanisms of sexual dimorphism from the development of germ cells to embryos. In past years, the YY super-male and XX neo-male yellow catfish were successfully created by sex reversal technology and identified by the robust sex-specific molecular markers [28,29,30,31], which is an excellent experimental model to explore the Y and X spermatozoa. In this study, we compared gene expression and genomic DNA methylation between X and Y spermatozoa (from XX neo-male and YY super-male, respectively) in yellow catfish. To further investigate the contributions of paternal DNA to embryonic development, we crossed XX neo-males with XX females to generate XX female fish, and YY super-males with XX females to generate XY male fish. Through a combination of SNP and transcriptomic dynamics analysis of XX female and XY male fish, we identified the MZT process and paternal activation from five early stages, which are 2-cell, 64-cell, high-blastocyst, low-blastocyst, and the bud of embryonic development. We revealed the key point of maternal degradation and zygotic as well as paternal activation. Besides, we found sex differences emerged from the bud stage, although at earlier stages XY males expressed more zygotic genes and indicated higher expression than XX females. Moreover, we detected the influence of paternal inheritance on sexual dimorphism of genome activation. Our studies provide clues for further elucidating the mechanisms under sex determination and differentiation of fish species.

## 2. Materials and Methods

### 2.1. Experimental Fish and Breeding

Two-year-old sexually mature YY super-male (142.99 ± 11.03 g) and XX female (74.74 ± 9.96 g) yellow catfish were obtained from our breeding center in Xiantao, Hubei, China. Due to technical obstacles, it is difficult to distinguish or isolate X/Y sperm from XY male fish. So, YY super-males are sources to obtain Y sperm, while XX neo-male fish are considered as good sources to isolate X sperm. In this research, XX neo-male yellow catfish were obtained through artificial sex reversal according to our previous studies [30,31,32]. They were also two years old (139.02 ± 11.72 g) and sexually mature when selected as experimental fish. All fish were maintained at 23–27 °C in a recirculating freshwater system on a 12 h light/12 h dark cycle. Water parameters, including dissolved oxygen (DO), nitrites (NO_2_-N), and ammonia (NH_4_-N) were maintained between 6.4 ± 1.5 mg/L, 0.02 ± 0.006 mg/L, and 0.3 ± 0.1 mg/L. All fish were fed with 40% crude protein commercial feed (from Hubei Jiakang Biotechnology Co., Ltd., Xiantao, China) twice a day (at 7:00 a.m. and 6:00 p.m.), respectively. XX and XY embryos were obtained from the crossing of XX female with XX neo-male, YY super-male yellow catfish separately. All experimental protocols in parental fish and embryos were in accordance with the principles approved by the Animal Experiment Ethical Inspection of the Laboratory Animal Center, Huazhong Agricultural University.

### 2.2. Preparation of Percoll Gradient-Centrifuged Sperm

Percoll gradient-centrifuged sperm were prepared according to the method of Chu et al. [33]. All experimental fish were fasted overnight and anesthetized with tricaine methanesulfonate 200 mg/L, followed by removal of the testes. Then, the gonad index (GSI) values of XX neo-male and YY male yellow catfish were calculated. Gaps were made with a scalpel in each seminiferous lobule and semen was squeezed out with a pair of forceps. The semen was washed in Krebs-Ringer bicarbonate medium (119.4 mM NaCl, 4.8 mM KCl, 1.7 mM CaCl_2_, 1.2 mM KH_2_PO_4_, 1.2 mM Mg_2_SO_4_, 25 mM NaHCO_3_, 1 mM sodium pyruvate, 25 mM sodium lactate, 5.6 mM glucose, 1 U/mL penicillin G, 1 g/mL streptomycin sulfate, and 28 M phenol red), supplemented with 0.3% bovine serum albumin (KRB-BSA) and centrifuged (650× *g*, 10 min).

To prepare Percoll gradient-centrifuged sperm, the sperm pellet, collected as described above, was diluted to 1 mL with KRB-Heps (consisting of the same components as KRB, except that 21 mM Hepes and 4 mM NaHCO_3_ were used in place of 25 mM NaHCO_3_) and centrifuged at room temperature through a two-step discontinuous Percoll gradient (45% and 90% KRB-Heps) at 650× *g* for 30 min. After Percoll gradient-centrifuging, fresh semen was divided into two layers. Immature sperm cannot enter 90% Percoll-gradient and were located at the interphase fraction between the two Percoll gradients, and mature sperm were distributed at the bottom as a pellet. The sperm pellets were washed once in KRB-Heps (650× *g*, 10 min) and incubated in KRB-BSA (30 min, 25 °C, 5% CO_2_), then centrifuged (650× *g*, 10 min) again. Percoll gradient-centrifuged sperm pellets were resuspended in KRB to a concentration of 2 × 10^7^ cells/uL, and a portion was used to analyze the sperm motility parameters. The remaining samples from the same yellow catfish were divided into two halves, then centrifuged (650× *g*, 10 min) and stored at −80 °C for RNA-seq and DNA methylation analysis.

### 2.3. RNA-seq of X/Y Sperm and Data Analysis

Total RNA was extracted from X and Y sperms by TRIzol^®^ Reagent (Invitrogen, Carlsbad, CA, USA/cat.15596-018) following the manufacturer’s instructions with an optional RNAse-free DNAse (Promega, Madison, WI, USA/cat.9001-99-4) digestion step included. The quantity and quality of RNA were determined by Nanodrop 2000 spectrophotometer (Thermos Scientific, Waltham, MA, USA) and 1% agarose gel electrophoresis. Only RNA samples of high quality were used for the cDNA synthesis. Poly (A) mRNA was purified by oligo (dT) magnetic beads and then was sheared into short fragments by adding a fragmentation buffer. First-strand cDNA was synthesized using random hexamer-primed reverse transcription and the second strand was then generated using components of dNTPs, buffer, DNA polymerase I, and RNase H. After purification, adapters were ligated to the cDNA, and then enriched using PCR. Library quality was assessed on the Agilent Bioanalyzer 2100 and qPCR. Three replicated portions of each sperm sample were subjected to RNA-seq. So, a total of six cDNA libraries were sequenced on an Illumina NovaSeq 6000 platform with 150 bp paired-end approach (by Novogene Co., Ltd., Beijing, China). Fastp (Version 0.20.0) [34] was used to trim the low-quality bases and the adapters. Reads were mapped to the yellow catfish (*Pelteobagrus fulvidraco*) reference genome using HISAT2 (Version 2.1.0) [35]. Transcript assembly and gene abundance estimation were performed using StringTie (Version 2.1.1). Three replicated samples were sequenced for X and Y sperm, respectively. Genes with mean FPKM > 1.0 in any group were retained for differential expression analysis using R package DESeq2 (Version 1.34.0) [36]. Thresholds of padj < 0.05 and |FoldChange| > 2.0 were used for filtering differentially expressed genes (DEGs). Gene Ontology (GO) analysis was performed by clusterProfiler (Version 4.2.0) [37]. Heatmaps with k-means clustering were used to display gene expression patterns over groups of samples.

### 2.4. WGBS of X/Y Sperm and Data Analysis

Sperm DNA isolation, bisulfite conversion, and sequencing were carried out by the Novogene company (https://en.novogene.com/ (accessed on 2 December 2022)). Three replicated samples were bisulfite-converted and sequenced. Lamda DNA spike-in was added to confirm the bisulfite conversion efficiency. The libraries with DNA fragment lengths of 100–500 bp were subjected to 150 bp pair-end sequencing on Illumina NovaSeq 6000. After sequencing, the raw reads were filtered by SOAPnuke (Version 1.5.5) (https://github.com/BGI-flexlab/SOAPnuke) to remove adaptor sequences, contamination, and low-quality reads. Then, the clean data were mapped to the yellow catfish (*Pelteobagrus fulvidraco*) genome by Bismark (Version 0.18.1) [38] with default options. Duplicated sequences were removed by deduplicate_bismark. Coverage depth was calculated by SAMtools and the methylation level of each cytosine was extracted by Bismark function “bismark_methylation_extractor” from BAM files. Methylpy (Version 1.4.0) [39] was used to call the methylation state for all Cs over the genome and convert the methylation level to bigwig files, which can be visualized by IGV or a similar browser.

Differentially methylated regions (DMRs) were identified by the R package methylKit (Version 1.20.0) [40] by thresholds of q value < 0.01 and methylation difference ≥25%. Genome-wide methylation distribution was indicated by density plot, and distribution of methylation difference was shown by histogram. The annotation of DMRs to gene elements was implemented by the R package ChIPseeker (Version 1.16.1) [41]. Violin plot was used to indicate CpG methylation levels at different regions of the genome. The average methylation profile was calculated for both 2 kb upstream and downstream from the center of each DMR in 50 bp bins and plotted by the plotProfile function in deepTools (Version 3.3.0) [42].

### 2.5. Identification of Maternal, Zygote, and Paternal SNPs and Genes in Early Embryonic Development through WGRS and RNA-seq

To detect maternal, zygotic, and paternal gene expression from the transcriptome, we adopted the hybrid SNP mapping method as mentioned by Harvey et al. (2013) [43]. The XX females were crossed with XX sex-reversed males (XX neo-male) and YY super-males, respectively (Figure 1A). The DNA of the parental fish was extracted for whole-genome re-sequencing (WGRS) by the Novogene company as mentioned above. Three replicated samples were sequenced. Eggs from XX females and embryos of each cross at the five early developmental stages, comprising 2-cell, 64-cell, high-blastocyst, low-blastocyst, and bud stages, were collected for mRNA sequencing (Figure 1A). Three replicates were also sequenced. The RNA extraction and sequencing strategies were the same as described above. The pipeline of identification of maternal, zygotic, and paternal expressed genes is shown in Figure 2. Whole-genome sequencing reads were mapped by Burrows-Wheeler Aligner (BWA, Version 0.7.17) [44]. Subsequent BAM files were processed with Picard. RNA-seq cleaned reads were quality controlled by FastQC (Version 0.11.9) [45], then mapped to the yellow catfish genome by STAR (Version 2.7.4a) [46] with default parameters. BAM files were sorted by Samtools (Version 1.13) [47]. SNPs were detected by GATK (Version 4.1.0.0) [48] for both parental fish and embryos, referring to the 1000 Genomes Project variant calling pipeline. Homozygous and distinct sites between parents were reserved for the identification of maternal and zygotic expressed genes in both crosses. In the transcriptome, genes harboring parental hybrid loci are regarded as zygotic genes, otherwise harboring homozygous maternal loci are considered as maternal genes, and those harboring only paternal loci are paternal genes. After the identification of maternal, zygotic, and paternal genes, we performed transcriptomic gene expression analysis using the same methods as used in the RNA-seq of X/Y sperm.

### 2.6. Statistics

The statistical analysis was carried out using GraphPad Prism (Version 8.0). Student’s two-tailed unpaired t-test was used for statistical comparisons and data are shown as mean ± SD. A *p* value of <0.05 was considered significant (*), while *p* < 0.01 and *p* < 0.001 were considered extremely significant, (**) and (***), respectively.

## 3. Results

### 3.1. Gene Expression Differences between X and Y Sperm of Yellow Catfish

GSI values of XX neo-male and YY super-male yellow catfish were 0.78 ± 0.06 and 0.79 ± 0.04 individually. To gain molecular differences between X and Y sperm generated from XX neo-male and YY super-male yellow catfish, we performed RNA-seq to detect the gene expression program (Figure 1A). Three biological replicates from each group were sequenced to generate a sufficient number of reads. We obtained an average high alignment rate to the genome (Appendix A). Sample correlation and principal component analysis (PCA) showed that biological replicates clustered together and the two sperm groups were clearly distinguished from each other (Appendix A). A total of 20,849 genes were detected from X sperm while 21,071 genes were detected from Y sperm, of which 20,123 genes were commonly expressed through overlapping of the two gene sets, along with 726 X-specifically and 948 Y-specifically expressed genes (Figure 1B). In order to investigate the relationship between the variation of gene expression and the increasing ratio of gene numbers, we inspected the cumulative distributions of genes in X and Y sperms (Figure 1C). We used different color curves to represent gene distributions of different sperms in the figure; the left side of the dotted vertical line represents the cumulative distribution of 90% genes, while the right side represents the cumulative distribution of the top 10% highly expressed genes. Generally, the cumulative distribution of highly expressed genes was very similar in both sperms with 90% of genes showing an expression level lower than 1.650 (log10 (FPKM)) in X sperm and 1.637 in Y sperm. However, when focusing on 70% of the total genes (log10 (FPKM) < 1.0), we observed that the blue curve lay below the red curve, which means that the overall gene expression in Y sperm was slightly higher. In addition, we identified 3054 differentially expressed genes (DEGs) from Y sperm vs. X sperm (Figure 1D,E). Interestingly, most of the DEGs (84.3%) were upregulated, and only a small proportion (15.7%) was downregulated (Appendix A).

Functional annotation of the upregulated DEGs revealed significant enrichment of cytokine–cytokine receptor interaction, TGF-beta signaling pathway, toll-like receptor signaling pathway, MAPK signaling pathway, JAK-STAT signaling pathway, TNF signaling pathway, Ras signaling pathway, antigen processing, and presentation and chemokine signaling pathway (Figure 1F). Among these pathways, cytokine–cytokine receptor interaction, TGF-beta, and toll-like receptor signaling are reported to play important roles in reproductive processes such as spermatogenesis, sperm capacitation, and fertilization [49]. Some pivotal genes on these pathways, such as *tgfb1/2* and *tlr5,* are upregulated from Y sperm vs. X sperm (Figure 1G). These genes and their family members are well known to play essential roles in spermatogenesis, male gonad development, and sex differentiation [49,50]. Their upregulation in Y sperm may suggest a potential difference between the two types of sperm. Cross-talk analysis indicated that the pathways interact or collaborate to affect germ cell migration and the development of primary sexual characteristics (Figure 1H).

### 3.2. Genome-Wide DNA Methylation Differences between X and Y Sperm of Yellow Catfish

To dissect the specificity of epigenetics of yellow catfish X and Y sperm and explore the driving factors underlying differential gene expression, we assessed the DNA methylation landscape of X and Y sperm by whole-genome bisulfite sequencing (WGBS) (Appendix A). Three biological replicates of each sperm type were sequenced and about 255 million clean reads were obtained on average. In addition, lambda DNA spike-in was added to confirm the bisulfite conversion efficiency and efficiently, and an average of 99.6% C (cytosine) was converted successfully. The average mapping rate to the yellow catfish genome was about 80%. Nearly 98% of Cs from CpG context were methylated, while only 0.5% Cs from CHG and 1.4% of Cs from CHH (where H is any base except G) were methylated (Appendix A). Due to the majority of Cs being in the CpG context and detected as methylated in the yellow catfish genome, we focused on the methylation level of global CpG context. We found that the methylation level in Y sperm was slightly lower than in X sperm (Figure 3A). About 37.1% of CpG sites over the genome in Y sperm showed methylation levels higher than those of X sperm, while 39.8% of CpG sites in X sperm showed methylation levels higher than those of Y sperm. Although a 2.7% difference is very small, it represents 468,468 CpG sites and thousands of genes over the genome.

To compare methylation differences at the chromosome level, elements level and gene level, we identified differentially methylated regions (DMRs) from Y vs. X sperm. DMRs were defined as regions with methylation differences higher than 25% and a q value (by SLM correction) lower than 0.01. We found 10,395 DMRs, including 4278 hyper and 6117 hypo regions, which locate at 2352 genes and 3247 genes, respectively (Figure 3B, Appendix A). The methylation difference was pronounced at the center and along ±2 kb of all DMRs (Figure 3B), with the majority of the DMRs being ~36% hyper or hypomethylated (Appendix A). DMRs were assigned to 26 chromosomes of yellow catfish, and the percentage of hyper and hypomethylated regions per chromosome is shown in Figure 3C. Almost at all chromosomes, the percentage of hypomethylated regions was higher than that of hypermethylated regions. To further study the distribution of methylated regions in the genome, DMRs were assigned to different genomic regions including promoter, exon, intron, downstream, and intergenic regions (Figure 3D). Through comparison of hyper and hypo-DMRs, we found more hyper-DMRs were assigned to promoter regions while more hypo-DMRs were assigned to distal intergenic regions, and the majority of DMRs were located in non-coding regions. The annotation of DMRs to genes showed that some genes contained both hypermethylated and hypomethylated regions. Through overlapping the hyper and hypo-DMR harboring genes, we found 1793 hyper-specific and 2688 hypo-specific genes (Figure 3E). The hypomethylated genes were numerically dominant in Y sperm, which is consistent with the gene expression program finding (from RNA-seq) that the upregulated genes are numerically dominant in Y sperm.

### 3.3. Dissection of DEGs Regulated by Methylation Modification in X/Y Sperm

To investigate whether methylation changes affect gene expression, we investigated the expression pattern of hyper and hypomethylated genes in X and Y sperm (Figure 4A,B). Theoretically, the hypermethylated genes should be downregulated and the hypomethylated genes should be upregulated (Y sperm vs. X sperm), but from each of the heatmaps, we only found one cluster that matched the theory. From cluster1 of Figure 3A and cluster3 of Figure 3B, we identified 613 genes that were hypermethylated and 1382 genes that were hypomethylated, respectively. Furthermore, we focused on significant DEGs from these hyper and hypomethylated genes (Figure 4C,D). From the Venn diagrams, 54 of the 613 hypermethylated and downregulated genes were DEGs, while 322 of the 1382 hypomethylated and upregulated genes were DEGs. Pathway enrichment analysis indicated that the 322 genes were significantly enriched at cytokine–cytokine receptor interaction, MAPK signaling pathway, JAK-STAT signaling pathway, and toll-like receptor signaling pathway. In order to understand how methylation change regulates gene expression, we focused on a group of genes that are differentially methylated at promoter regions (Appendix A). In this study, we found some essential genes such as *crlf1*, *tnfrsf*, *il1b* from the cytokine–cytokine receptor interaction pathway were upregulated and hypomethylated at promoters (Figure 4E), which indicates this pathway is potentially regulated by methylation modification.

### 3.4. Identification and Profiling of Maternal-to-Zygotic Transition (MZT) of XX and XY Yellow Catfish during Early Embryonic Development

The initiation of animal development is guided by maternal gene products; then, the control of development is handed to the zygotic genome during the process of MZT [51]. To trace the maternal degradation and determine the zygotic activation starting time in yellow catfish, we studied five early stages of embryonic development to identify maternal and zygote genes and explore their transcriptome dynamics. We performed whole genome re-sequencing (WGRS) for the parental yellow catfish and RNA sequencing (RNA-seq) for the five embryonic stages of 2-cell, 64-cell, high-blastocyst, low-blastocyst, and bud (Figure 1A and Appendix A). Three biological replicates were sequenced for each stage. The WGRS genomes between biological replicates were very similar (about 98% similarity) (Appendix A) and the RNA-seq samples were also highly correlated between biological replicates (Appendix A). SNPs (single nucleotide polymorphisms) called from parents were used as molecular markers to identify maternal and zygotic genes during embryonic development (Figure 2). Generally, a gene locus is homozygous in any sexes of parents, but difference between the sexes could be identified as an SNP site. In total, from WGRS of the XX neo-male and XX female parents we identified 100,477 parental SNPs, while from YY super-male and XX female parents we identified 119,193 parental SNPs, corresponding to 13,093 and 13,690 genes, respectively. Then, these parental SNPs were taken as markers to identify maternal and zygotic SNPs from RNA-seq data of the XX female offspring (from the crossing of XX neo-males and XX females) and XY male offspring (from the crossing of YY super-males and XX females), respectively. For the embryo, the maternal gene is defined as a gene containing a homozygous locus that is consistent with the maternal genotype, while a zygote gene contains a heterozygous site and the genotype is a combination of the father and mother. The number of SNPs identified from the five stages is shown in Appendix A and the corresponding numbers of assigned genes are shown in Figure 5A. Although from both XX and XY embryos we observed similar dynamics in gene numbers during the development, differences between sexes still potentially exist and need to be explored.

According to the MTZ analysis, we observed a rapid decline in the number of maternal genes and a great increase in the number of zygote genes in the latter two stages, the low-blastocyst and the bud stages (Figure 5A). This observation indicates that the transition from high-blastocyst to low-blastocyst is a pivotal time point of MTZ. Then we screened out the maternal degradation genes and zygotic activation genes from the comparison of gene expression of these two stages. From XX female embryos we identified 703 maternal and 1068 zygote genes, while from XY male embryo we identified 652 maternal and 1264 zygote genes (Appendix A, Figure 5B). From both XX and XY embryos, there were more zygotic activation genes than maternal degradation genes, which implies the initiation of development is handed to the zygotic genome from maternal maintenance. Through overlapping maternal genes as well as zygotic genes between XX female embryos and XY male embryos, we found the XX female embryo has more specific maternal degradation genes (472 vs. 421), while the XY male embryo has more specific zygotic activation genes (735 vs. 539) (Figure 5B). This result suggests that the maternal degradation is comparable between the two sexes of embryos, but the initiation of zygotic development is more active in the XY male embryo. When focusing on the function of these genes, the maternal genes are mainly enriched in catabolic and metabolic activities of cells in both sexes of embryos (Figure 5C,D). In contrast, the zygotic genes are primarily involved in the development of organs (Figure 5E,F). Although in both sexes the zygotic genes shared most of the functions, XY males but not XX females developed sexual characteristics (Figure 5F).

### 3.5. Comparative Analysis of Sex Determination Genes and PGC-Related Genes of XX and XY Embryos during the Maternal-to-Zygotic Transition

To further investigate whether sex instruction begins from the early embryo development before gonad formation and if differences exist between sexes on maternal genome degradation and zygotic genome activation, we focused on the expression dynamics of a group of sex determination genes including *dmrts*, *foxl2,* and *cyp19a1*. Unexpectedly, we observed little difference between XX female and XY male yellow catfish in the expression pattern of these genes (Appendix A), probably because at the observation time it was too early to see differences in these genes. However, primordial germ cells (PGCs) are more sensitive if seeking to indicate the differences between sexes at the early developmental stages. PGC is well known to be the first germ cell population established during early embryonic development and acts as a precursor for both the oocytes and spermatogonia. Besides, PGCs are directly related to gonadal formation and sex differentiation [52,53]. To detect sexual differences from PGCs, firstly we examined the PGC self-renewal genes *sox2* and *nanog*, and the PGC formation and migration genes *ddx4*, *dnd1,* and *nanos3* from early stages. However, XX female and XY male fish demonstrated similar expression dynamics in these genes (Figure 6A). Then, we screened out 287 and 263 PGC-related genes from maternal genes of XX females and XY males separately through Spearman correlation analysis with a threshold of r > 0.9 and *p* value < 0.05, and found the numbers were 532 and 623 from zygotic activation genes (Figure 6B, Appendix A). We also focused on the proportions of these PGC-related genes in maternal degradation and zygotic activation (Figure 6C). The results indicated that an equal proportion (46%) of XX-specific and XY-specific maternal genes is PGC-related, while a similar proportion (51% vs. 50%) of XX-specific and XY-specific zygotic activation genes are PGC-related genes. However, a higher proportion of PGC-related genes in zygotic activating than in maternal degradation was observed (48% vs. 30%), implying that the PGCs begin to be activated during the MZT process. To investigate if PGC-related genes changed more than non-PGC-related genes in gene expression during MZT, we examined gene expression changes from the comparison of the high-blastocyst and low-blastocyst stages. Genes with fold changes higher than two (|log2FC| ≥ 1.0) are regarded as relatively highly variable genes. The proportions of highly variable genes (colored in red and green, respectively) were examined and compared for sex, maternal, and zygotic properties, as well as between PGC and non-PGC groups, respectively (Figure 6D). The results showed that in both XX female and XY male yellow catfish, the maternal PGC and non-PGC groups had similar proportions of sharply decreased genes (the red colored). In comparison, the zygotic PGC groups exhibited much higher proportions of rapidly increased genes than non-PGC groups. When comparing the zygotic PGCs between sexes, the XY male fish showed a higher proportion of rapidly increased genes (53.7% vs. 49.9%), suggesting an earlier development initiated in male yellow catfish. Function analysis of these rapidly increased genes from XY male fish showed enrichment for terms related to male gonad development and male sex differentiation. Genes of these terms include *smad4*, *h3f3a*, *gata6*, *lrrc6*, *ube3a*, *eif2s2*, *foxo3*, *tgfbr1*, *fancf*, *bcl2l1,* and *wnt4*, of which *gata6* is activated at the low-blastocyst stage and differentially expressed at bud stage between sexes, with a much higher FPKM value in XX female yellow catfish. We focused on *gata6* because it is a crucial transcript factor involved in embryonic development. In addition, the expression level increased at the blastula stage, demonstrating its significance in morphogenesis. A previous study indicated that in Japanese flounder, *gata6* participated in gonadogenesis, regulation of gender-related genes, estrogen formation, and gonadal function maintenance [54].

In addition, from global gene expression analysis, we found that *ddx5*, a member of the DEAD-box gene family, also decreased gradually at the first three stages and then increased sharply at the bud stage. Previous study in mice has shown that *ddx5*, as a transcription cofactor, plays a variety of functions in undifferentiated spermatogonial cells, such as participating in the maintenance of the germline, regulating the expression of genes involved in the cell cycle, and maintaining correct gene splicing during spermatogenesis [55]. Although its function in fish is unclear at present, a similar member *ddx4* (*vasa*) from the same family is well studied. *Vasa* is known to be a marker of PGC and is usually used to track PGC initiation, migration, gonad development, and sex differentiation [56]. Its expression can affect the number of PGCs, and thus affect gonadal development and the sex differentiation [18]. Interestingly, in this study, *vasa* showed almost no difference in the early development of yellow catfish, while *ddx5* was differentially expressed between sexes at the bud stage, an initiation time of sex determination. It is reasonable to speculate that temporal and spatial differences exist between *ddx5* and *vasa* expression. Compared with *vasa*, the higher expression of *ddx5* in XY males after the key transition point demonstrates that it plays a more important role in male gonad development and spermatogenesis.

Generally, *gata6* and *ddx5* are two newly discovered genes which are potentially related to gonad development in yellow catfish. Their sexually dimorphic expression at the bud stage might influence sex determination and differentiation during the early embryonic development.

### 3.6. Detecting Influence of Paternal Inheritance on Sexual Dimorphism Genome Activation of Yellow Catfish during Embryonic Development

X and Y sperm play essential roles in the formation of sexual dimorphism and highly specialized developmental functions. However, in the early stages of embryonic development of yellow catfish, the contributions of the sperm to development and sexual dimorphism are still unclear. Through the identification of paternal SNPs and related genes, especially sex-specific paternal functions, we assessed the contributions of paternal inheritance of sperm DNA on sexual dimorphism from five early embryonic stages. Although maternal degradation and zygotic activation are dominant activities in the early stages, paternal gene activation also occurs during the development and shows the same trend as zygotic activation (Figure 7A). Paternal activation seems to be closely related to zygotic activation because of the strong correlation (Pearson correlation r = 0.99). From high-blastocyst to low-blastocyst, there are rapid increases in the number of paternal SNPs in both sexes (Figure 7B). Significant differences between sexes are detected at 2-cell and 64-cell stages, then diminished at the key point of MZT, and reappear at the low-blastocyst and bud stages. At all stages, XY male fish showed a higher percentage of paternal SNP than XX female fish. Accordingly, sex-specific paternal genes increased significantly at the low-blastocyst and bud stages (Figure 7C), indicating a potential activation of paternal development. When focusing on sex-related genes in the last two stages, XY fish expressed more male gonad developmental genes while XX fish expressed more female gonad developmental and sex differentiation genes (Figure 7D). Function analysis of sex-specific genes indicates that XX fish are enriched via female-related terms such as the TNF signaling pathway, oocyte meiosis, and estrogen hormone synthesis, while XY male fish are enriched via toll, MAPK, and chemokine signaling pathways (Figure 7E), which is consistent with the previous transcriptomic analysis results of X and Y sperm.

## 4. Discussion

Sexual body size dimorphism has been observed in yellow catfish males that grow faster than females [32]. This morphological difference between sexes prompted us to reveal mechanisms that control sex determination during embryonic development. Due to the fact that sex determination is usually controlled by a combination of genetic and environmental influences and is highly plastic [57], we were able to create YY super-male and XX neo-male yellow catfish through sex reversal technology and genetic technology [28,30,31]. The Y and X sperm generated from these male fish and the offspring of hybrid and X oocytes were excellent materials with which to detect sexual differences from early embryonic development.

Through comparative transcriptome analysis of the X and Y sperm, we found most of the DEGs were upregulated in Y sperm while only a small proportion was downregulated, but no difference was observed in morphology and motility. This finding is supported by recent studies that found the only difference lies in sperm DNA content, and negligible or no differences between X and Y sperm in morphology, physical, or chemical properties observed [4,5,7]. We found several upregulated pathways in Y sperm that contributed to germ cell migration and primary sexual development. In specific, toll-like, TGF-beta, and cytokine–cytokine pathways are well known to be involved in the spermatogenesis [49]. Toll-like receptor family genes are reported to be expressed in mammalian sperm and impair fertilization by altering sperm capacitation, so their differential expression might confer functional differences between X-sperm and Y-sperm [50]. In mice, Tlr7/8 is expressed in X-sperm but not the Y-sperm; incubating sperm with Tlr7/8 ligands can dramatically suppress the hyperactivated motility of X-sperm, which allows the successful separation of X-sperm from Y-sperm [50]. In this study, we investigated the toll family but found almost no expression of *tlr7/8* in yellow catfish, whereas *tlr5* is a differential expressed member that is significantly higher in Y sperm than X sperm. Functionally, this fish-specific member plays an important role in innate immunity [58,59]. The TGF-beta superfamily contains at least 40 members, many of which are produced in the mammalian testes to facilitate the formation of the sperm [60]. NODAL is also a spermatogenesis-related gene that is involved in the regulation of gonocyte differentiation and fetal testis development in the human [61]. We found two members of the TGF-beta family, *tgfb1* and *tgfb2*, along with *nodal*, participated in the TGF-beta signaling pathway and cytokine–cytokine receptor interactions. These signaling pathways are combined to form a complex regulatory network for spermatogenesis in Y sperm.

DNA methylation is a well-studied epigenetic modification essential to mammalian development. It plays a significant role in gene silencing, protection against spurious repetitive element activity, genomic stability during mitosis, and parental imprinting [62]. In this study, from the methylation results of sperm, we identified more hypomethylated than hypermethylated genes from Y vs. X sperm (2512 vs. 1653). However, only 37.1% (613 of 1653) of the hypermethylated genes were downregulated, while 55% (1382 of 2512) of the hypomethylated genes were upregulated, which means that the upregulated DEGs are more affected by methylation change. Large proportions of DEGs are not affected by methylation change, which suggests that these genes may be regulated by other factors, such as chromatin remodelers, enhancers, transposable elements, or other modifications on chromatins including but not limited to acetylation, phosphorylation, and citrullination [63]. For some genes, their promoter regions are differentially methylated between X and Y sperm. The promoter is a very important component that controls gene expression with methylation-like “switches” that determine the activity of gene promoters. Actually, the promoter itself does not control gene activity, but the methylation or demethylation of promoters can regulate gene expression by preventing or enabling transcription factors (activators or repressors) from binding to the regions [64]. In this study, 322 upregulated genes hypomethylated in Y sperm (Figure 3D), such as *crlf1*, *tnfrsf,* and *il1b,* were enriched at the cytokine–cytokine receptor interaction pathway (Figure 3E). From our cross-talk analysis, this pathway interacts with TGF-beta, toll-like, TNF signaling pathways, and is indirectly linked with germ cell migration as well as the development of primary sexual characteristics. The hypomethylation at the promoters resulted in the upregulation of the three genes, but whether they systematically regulated gene expression in that network needs to be further investigated.

During the early embryonic development, we investigated some sex-determining genes reported as markers in fish sex determination and differentiation, such as *cyp19a1*, *foxl2*, *dmrt1*, and *wnt4*, are widely reported sexual dimorphic genes in the teleost [65]. Among these, *cyp19a1*, *foxl2*, and *wnt4* are widely reported to be specific in ovary differentiation and *dmrt1* is involved in the testicular differentiation. We compared the expression of these genes at different stages of early development [14,65,66], from sperm to bud formation. Unexpectedly, we observed almost no difference in expression of these genes between the two sexes of yellow catfish; what is more, the average FPKM of *cyp19a1* was too low to indicate the expression, while *foxl2* and *wnt4* were only effectively expressed at the bud stage but no other stages. We speculate that it is too early to detect the expression of sexual markers before bud, due to the testicles and ovaries not yet being developed at this early time. From the combination of SNP and gene expression analysis, we identified the low-blastocyst stage as the transition point of MZT process and paternal activation. However, when comparing the expression of maternal, zygotic, and paternal genes between XX and XY yellow catfish, few differences were screened out until the bud stage. Thus, it seems that the markers of sex determination, sex differentiation, and PGCs begin to express from the bud stage. *Gata6* and *ddx5* are two newly identified genes that are differentially expressed at the bud stage between sexes in yellow catfish. They are reported to be involved in gonad development and have the potency to influence sex determination and differentiation during early embryonic development [55,67]. Additionally, *gata6* is a highly-conserved transcription factor and plays an important role in gonadal cell proliferation, differentiation, and endoderm development [54]. *Gata6* is expressed in the ovary during embryonic development and is crucial for follicle assembly, granulosa cell differentiation, postnatal follicle growth, and the luteinization [68], while Ddx5 is expressed by spermatogonia in mice and acts as a transcriptional co-activator in the germline maintenance [55]. In zebrafish, *ddx5* was expressed in developing gonads, and it is dispensable for testes development but essential for female sex differentiation and oocyte maturation [69]. In this study, *ddx5* was differentially expressed at the bud stage with much higher expression in XX females, which suggests that this gene is more important to the development of female yellow catfish. Nevertheless, the dimorphic function of these genes needs to be further revealed.

## 5. Conclusions

Overall, through comparative transcriptome analysis of X and Y sperm of yellow catfish, we found the majority of DEGs are upregulated in Y sperm. Function enrichment revealed that cytokine–cytokine interaction, toll-like and TGF-beta pathways, and crucial genes such as *tlr5*, *tgfb,* and *nodal* are correlated with sexual characteristic development and germ cell migration. Whole-genome DNA methylation of X and Y sperm indicated that most of DMRs are hypomethylated in Y sperm, suggesting genes of Y sperm are more active than X sperm, which is consistent with the overall transcriptomic gene expression being globally higher in Y sperm. Some cytokine pathway genes are directly demethylated at promoter regions and increased in expression. Through combination analysis of WGRS of parental yellow catfish and RNA-seq of five stages of early embryonic development, we identified the low-blastocyst stage as the critical transition point of MZT and paternal genome activation. Differences between XX female and XY male yellow catfish emerged at the bud stage but not at the earlier stages. *Gata6* and *ddx5* are potentially involved in early sex determination before gonad formation. These findings provide insights that reveal the mechanisms of sex determination and sexually dimorphic differentiation of yellow catfish.

## Figures and Tables

**Figure 1 biology-11-01818-f001:**
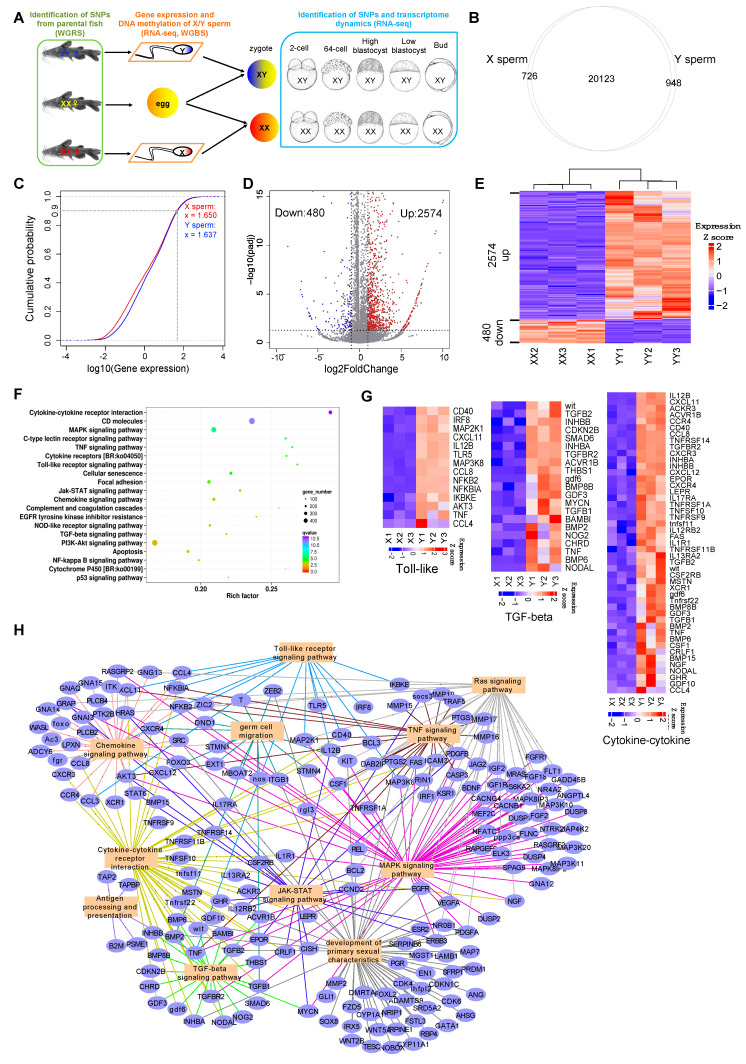
Overall design and gene expression summary of X sperm and Y sperm experiment. (**A**) Parental fish, generation of gametes and zygote, and sequencing strategies for fish, gametes, and embryos. In this study, we used WGRS for XX neo-males, XX females and YY super-males, RNA-seq and WGBS for X and Y sperm generated from the XX neo-males and YY super-males, and RNA-seq for XX female and XY male embryos from the development of 2-cell to bud stage. (**B**) Overlapping of genes expressed in X sperm and Y sperm. (**C**) The cumulative distribution of genes in X sperm and Y sperm. The x-axis indicates the expression level of genes and the y-axis indicates the cumulative percentage of genes. From left to right, gene expression level increases gradually, while from bottom to top the cumulative percentage of genes increases gradually. (**D**) Volcano plot of differentially expressed genes (DEGs). Thresholds for DEGs are Log2FoldChange ≥ 1.0 and padj ≤ 0.05. Red dotted: upregulated; blue dotted: downregulated. (**E**) Heatmap of DEG expression pattern. (**F**) Bubble plot of GO terms of up-DEGs. (**G**) Small heatmaps of selected pathways that related to sexual characteristics and spermatogenesis. (**H**) Cross talk of pathways and the collective effects on germ cell migration and development of primary sexual characteristics.

**Figure 2 biology-11-01818-f002:**
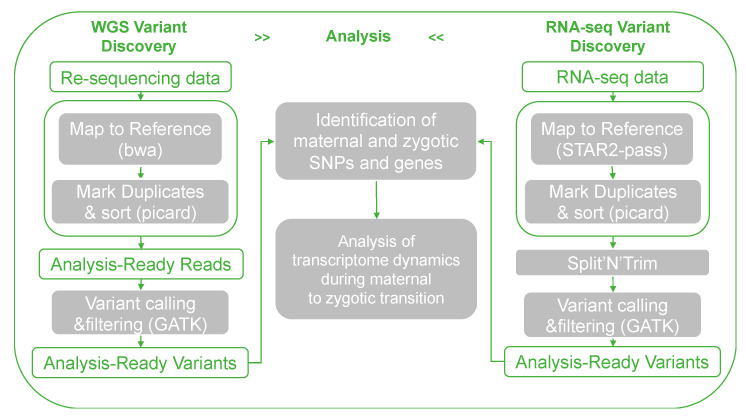
Pipeline for identification of maternal and zygotic SNPs and genes.

**Figure 3 biology-11-01818-f003:**
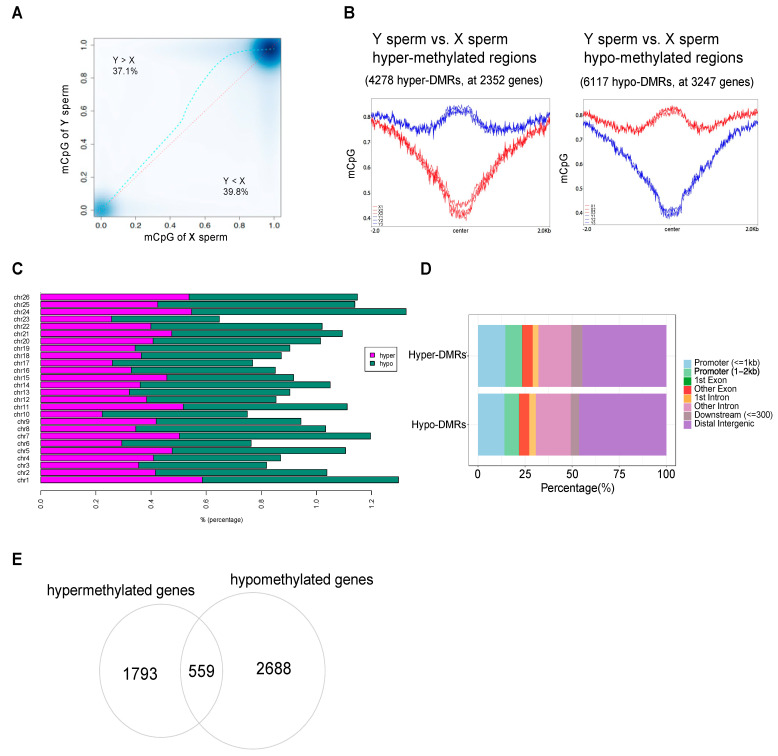
Genome-wide analysis of DNA methylation in X sperm and Y sperm. (**A**) Genome-wide hyper and hypomethylated CpG sites (Y sperm vs. X sperm). The red line represents the best fitting of linear regression, and the blue line is the smooth curve (Lowess method). (**B**) Average CpG methylation levels over ±2 kb of all differentially methylated regions (DMRs) from Y sperm vs. X sperm. Left: hypermethylated regions; right: hypomethylated regions. Red: X sperm; blue: Y sperm. The thresholds for filtering of DMRs: methylation difference ≥25% and q value < 0.01. (**C**) Percentage of hyper and hypomethylated regions per chromosome. (**D**) Assignment of DMRs to different genomic regions. (**E**) Overlap of hypermethylated genes with hypomethylated genes, and their specific genes.

**Figure 4 biology-11-01818-f004:**
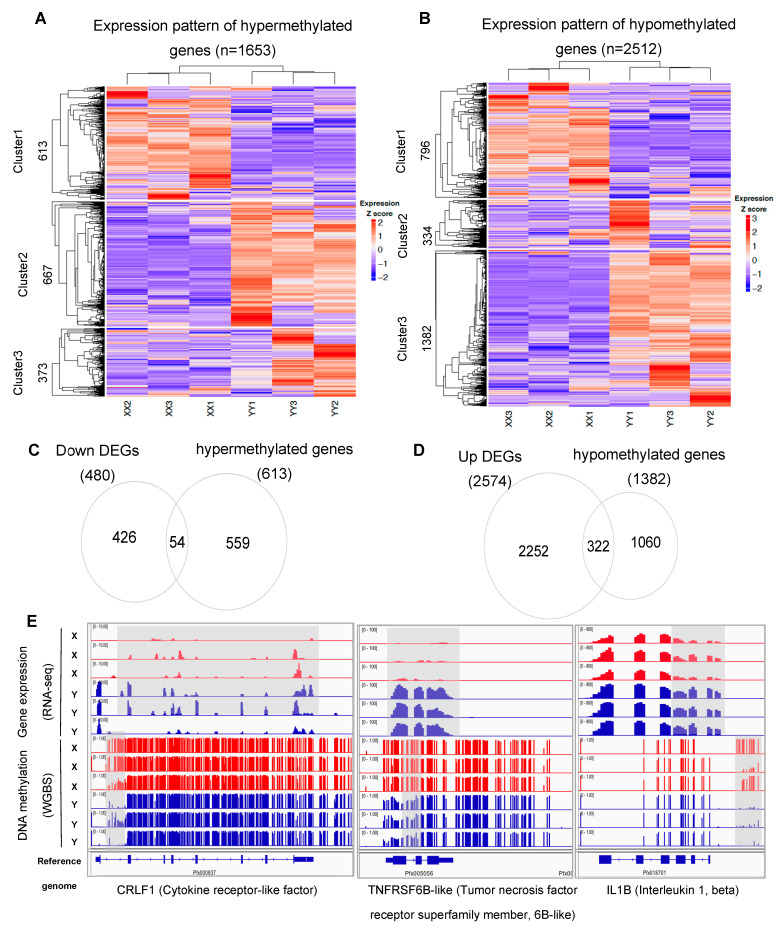
Integration of methylation with gene expression. (**A**) Expression clustering of hypermethylated genes (n = 1653, filtered by expression FPKM >0 in any of the samples) (left) and hypomethylated genes (n = 2512) (right) by k-means. (**B**) Overlap of downregulated genes with hypermethylated genes. (**C**) Overlap of downregulated genes with hypermethylated genes. (**D**) Overlap of upregulated genes with hypomethylated genes. (**E**) IGV visualization for CpG methylation of genes from cytokine–cytokine interaction pathway, and the corresponding gene expression levels (RNA-seq tracks). Blue: Y sperm; red: X sperm.

**Figure 5 biology-11-01818-f005:**
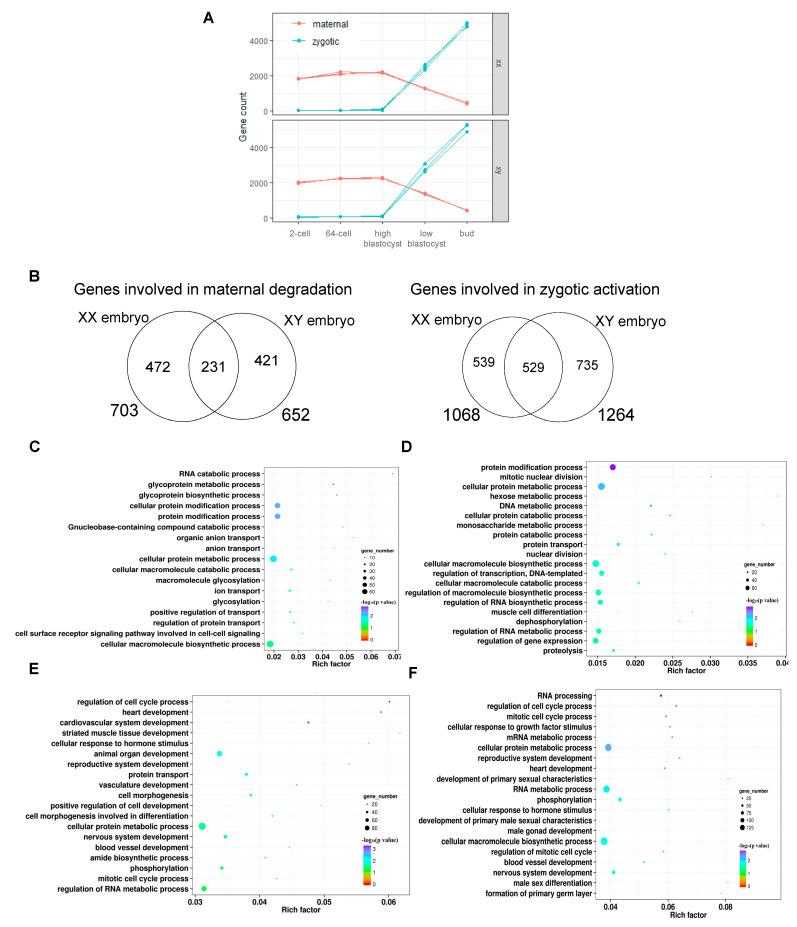
Identification of maternal and zygotic SNPs and genes from maternal-to-zygotic transition. (**A**) Maternal and zygotic expressed genes were detected at the low-blastocyst stage of both XX all-female and XY all-male yellow catfish. (**B**) Overlap of maternal degradation genes between XX all-female and XY all-male yellow catfish (left), and overlap of zygotic activation genes between XX all-female and XY all-male yellow catfish (right). (**C**) Function enrichment of maternal degradation genes from XX all-female. (**D**) Function enrichment of maternal degradation genes from XY all-male. (**E**) Function enrichment of zygotic activation genes from XX all-female. (**F**) Function enrichment of zygotic activation genes from XY all-male.

**Figure 6 biology-11-01818-f006:**
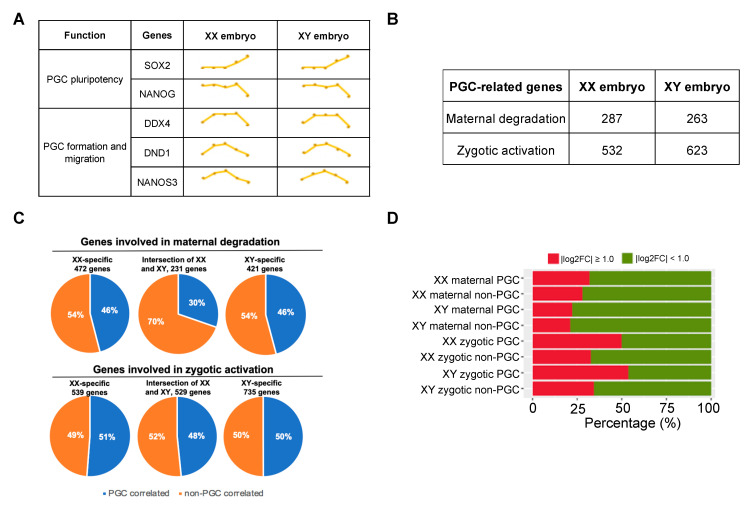
Identification of PGC (primordial germ cell)-related genes from maternal-to-zygotic transition. (**A**) The expression of PGC marker genes during early embryonic development of yellow catfish. (**B**) Statistics of PGC-correlated genes in maternal degradation genes and zygotic activation genes. (**C**) The proportion of PGC-correlated genes in XX-specific, XY-specific, the intersection of XX and XY, maternal degradation genes, and zygotic activation genes. (**D**) Expression fold change (|log2FC|) of maternal and zygotic PGC-correlated genes between the high-blastocyst and low-blastocyst stages. Red: percentage of genes with |log2FC| ≥ 1.0; green: |log2FC| < 1.0.

**Figure 7 biology-11-01818-f007:**
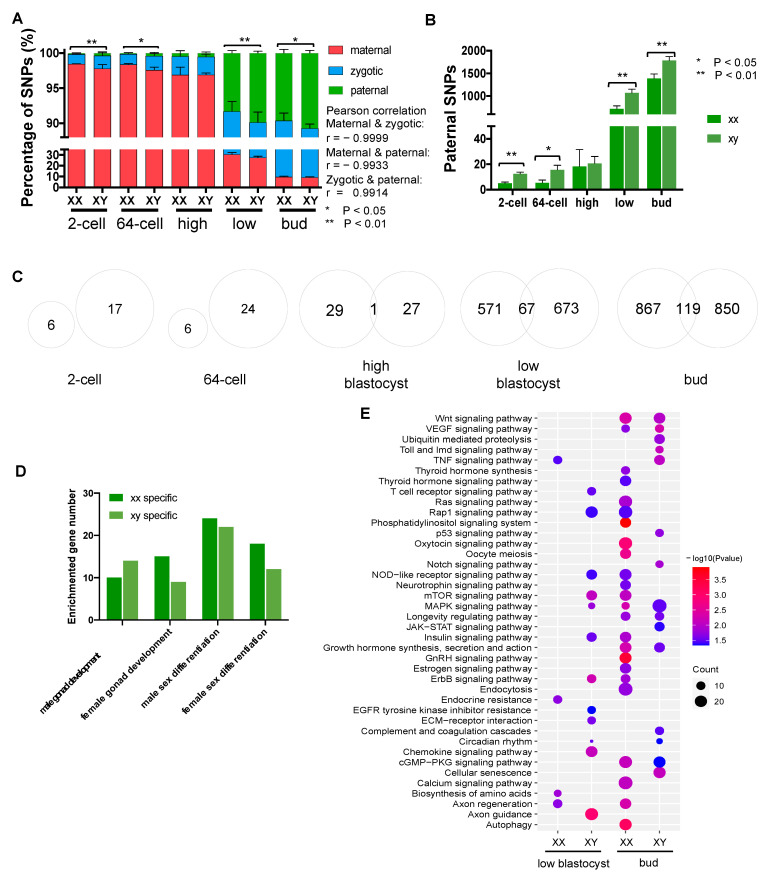
Dissecting contributions of paternal inheritance to sexual dimorphism of yellow catfish through identification of paternal SNPs and related genes from embryonic development. (**A**) Composition of maternal, zygotic, and paternal SNPs (by percentage) at each stage of embryonic development. (**B**) Paternal SNPs of female and male embryos at each stage. (**C**) Overlap of paternal genes at each stage. (**D**) Gonad development and sex differentiation genes in female and male embryos. (**E**) Pathways of female and male-specific paternal genes enriched in different sexes at low-blastocyst and bud stages are grouped respectively.

## Data Availability

The datasets generated from the current study are available in the NCBI SRA repository, under project number PRJNA830929 (https://www.ncbi.nlm.nih.gov/sra/?term=PRJNA830929 (accessed on 2 December 2022)).

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
