# Peer review of "Sexually Dimorphic Gene Expression in X and Y Sperms Instructs Sexual Dimorphism of Embryonic Genome Activation in Yellow Catfish (Pelteobagrus fulvidraco)"

_biology, 2022, doi:10.3390/biology11121818_

Round 1

Reviewer 1 Report

In this manuscript, Yang Xiong et al investigated the effects of paternal inheritance on sexual dimorphism of embryonic genome activation using yellow catfish. The study was carefully designed and executed. They performed sex-controlling breeding of yellow catfish and successfully obtained purified X and Y spermatozoa from XX neo-male and YY super-males. And found gene expression level is higher in Y sperm while the genome-wide DNA methylation level is slightly lower in Y sperm. Through embryonic development research, they revealed the key time point of maternal-to-zygotic transition (MZT) and found paternal genes were activated along with the zygotic gene activation. Influences of X and Y sperm on the sexual dimorphism of offspring were evaluated by a combined analysis of the parental genome and SNPs of five stages of embryos. Besides, they screened out gata6 and ddx5 as potential instructor genes for early sex determination and gonad development in yellow catfish. The study provided plenty of data and results to support their findings. The methods are valid and the findings are interesting. But there are several minor issues I'd like the authors to address:

1. Page 11, line 238. You mentioned key genes tgfb1/2 and tlr5 are up-regulated in Y sperm, are they differentially expressed in the process of embryonic development? Or, are they contributing to sexual dimorphism?

2. Page 11, line 243. In Figure 1H the characters are too small to be recognized.

3. Page 22-23, lines 429-449. You identified gata6 and ddx5 as potential instructors for early sex determination and gonad development in yellow catfish. They are differentially expressed at the bud stage. Did you investigate their expression in the X and Y sperm? Is the difference inherited from the gametes?

Reviewer 2 Report

In summary, this manuscript investigates differences between X and Y sperm, including gene expression, genomic methylation levels, regulation of methylation, and differential development of XX and XY zygotes. These results provide clues for the study of early embryonic differentiation and sex differentiation caused by X and Y sperms in yellow catfish. However, there are some concerns in the manuscript that require further explanation and description

1.     Kindly indicate the company name together with their cities of all the reagent and kits used in the paper. For example, in the part 2.3 RNA-seq, please state the company if RNA-seq is done by the company; and also please state the model of the instrument if the sequencing is done by yourself.

2.    The authors do not put the biological repetition in the materials and methods, however, it appears in the results section. Was it the mixing of 3 individuals and the parallel sequencing of 3 individuals, so a total of 9 individual tissues were used for the assay? The authors need to add the description in section of the materials and methods.

3.    Line 105, replace “artificially” by “artificial”.

4.    Line 107, Guide for the Care and Use of Laboratory Animals, Whether it is a reference? Please provide the corresponding literature.

5.    Line 108, “Before embryonic development ……”, “For embryonic development ……” seems much better.

6.    Line 112, add a detailed information about the Percoll gradient-centrifuged sperm, sperm pellets.

7.    In the result part, the authors use excessively long paragraphs to describe the result, such as Lines 212-247. In fact, dividing this complex paragraph into multiple paragraphs is more conducive to description and reading

8.    Lines 235-238, “cytokine-cytokine receptor interaction, TGF-beta signaling pathway, Toll-like receptor signaling pathway, MAPK signaling pathway, JAK-STAT signaling pathway, TNF signaling pathway, Ras signaling pathway, antigen processing and presentation and chemokine signaling pathway”, this sentence is mentioned again below, and the authors had told us the function of “cytokine-cytokine receptor interaction, TGF-beta signaling pathway, Toll-like receptor signaling pathway”, but the function of “MAPK signaling pathway, JAK-STAT signaling pathway, TNF signaling pathway, Ras signaling pathway, antigen processing and presentation and chemokine signaling pathway” is not mentioned, it confused me.

9.    Line 245, “suggest a potential difference between the ……”, A potential difference of what?

10. Figure 1 (H) , The authors said that cross-talk analysis indicates that these pathways work together on germ cell migration and primary sexual characteristics, but it is difficult to see how these pathways implying germ cell migration and primary sexual characteristics. Are there any key genes in the figure?

11. Lines 325-328, These two sentences should move to the discussion part, furthermore the function of promoters, as a common sense of molecular biology, does not need an explanation

12. Figure 4 (B), The authors said that low blastocyst stage is an important period of paternal activation, so zygotic gene expression should be similar before this period. However, from the figure, there are actually slight differences between XX and XY stained individuals from the 2-cell stage to the high blastocyst stage. How does the author explain this phenomenon?

13. Lines 450-458, These sentences seem like the description of the discussion section.

Reviewer 3 Report

After careful review, I am recommending this article for major revision before reconsidering for publication in the journal Biology

Comments are as follows.

Simple Summary:

1. line 19, grammatic error

2. line 25, is not clear 

3. overall sentence structure is poor and needs refinement and grammatical checking.

Abstract:

1. line 30, how can you check the sexes of the embryo before sex determination?

2. line 34, 

3. similar to the sample summary, the authors need to emphasize sentence structure and articulate sentences more. Many sentences appear like speculation rather than presenting results.

Introduction:

1. Lines 48-52, very poor sentence structure

2. Line 55, same as above not clear what the authors are trying to convey

3. Line 56-57, is that the objective of your study? 

4. The Second paragraph is not just like pieces of information together and nothing is put in context. Very poor organization.

5. Lines 77-78, are not clear, then what is the importance of X and y sperms?  

6. Overall, the introduction lacks organization, it's not adequate, putting random information and not putting things in context. Needs major refinement and additional information. 

Materials and methods

1. lacks critical information including age and physical parameters of the experimental model

2. How did the authors distinguish or isolate X and Y sperm?

3. Line 108 ?

4. Line 107, animal user protocol details?

5.  Did the authors have any information about GSI?

6. Line 127-128, for what?

7.  Line 133-135, irrelevant as mentioned earlier (correct it in the previous section)

8. How did quantify the RNA before cDNA synthesis? 

9. Is AGE enough to assess the quality of the RNA? if so how did the authors justify organic solvent/DNA contamination?

10.  Reagent information is lacking (manufacturer, country/cat. no)

11. What is the test used for normality and is the data corrected for it (if it needs)?

Results:

1. Section 3.1 is well written, however, better to avoid jargon.

2. Authors used 3 biological replicates to asses sperm characteristics, did they identify any differences between biological replicates?

3. What are the criteria the authors used to select biological replicates and what are the precautions they employed to avoid variation within the replicates?

4.  The authors have to explain the importance of introducing methylation of DNA well in advance before measuring it.

5. Lines 313-314, reference?

6.  Did the authors consider any other potential early sex determination genes other than the one mentioned in section 3.5?

Discussion

1.  Line 525, the genes mentioned here are having a role in spermatogenesis (according to the authors), then what is their importance in early sex determination in fish?

2. The authors should explain the importance of the methylation of DNA and how it is important in development. 

3. How do the authors conclude the transcript data is a reflection of what is going on in the system (protein, downstream targets)?

4. Finally, similar to the introduction, the discussion is short and inadequate.
